# Characterization and Antibacterial Activities of Carboxymethylated Paramylon from *Euglena gracilis*

**DOI:** 10.3390/polym14153022

**Published:** 2022-07-26

**Authors:** Liwei Gao, Xinjie Zhao, Meng Liu, Xiangzhong Zhao

**Affiliations:** School of Food Sciences and Engineering, Qilu University of Technology (Shandong Academy of Sciences), Jinan 250353, China; glw626816@163.com (L.G.); zhaoxinjie1107@163.com (X.Z.); 15864530661@163.com (M.L.)

**Keywords:** paramylon, carboxymethylated modification, water solubility, antibacterial activity

## Abstract

Paramylon from *Euglena gracilis* (EGP) is a polymeric polysaccharide composed of linear β-1,3 glucan. EGP has been proved to have antibacterial activity, but its effect is weak due to its water insolubility and high crystallinity. In order to change this deficiency, this experiment carried out carboxymethylated modification of EGP. Three carboxymethylated derivatives, C-EGP1, C-EGP2, and C-EGP3, with a degree of substitution (DS) of 0.14, 0.55, and 0.78, respectively, were synthesized by varying reaction conditions, such as the mass of chloroacetic acid and temperature. Fourier transform infrared spectroscopy (FTIR), gel permeation chromatography (GPC), and nuclear magnetic resonance (NMR) analysis confirmed the success of the carboxymethylated modification. The Congo red (CR) experiment, scanning electron microscopy (SEM), X-ray diffraction (XRD), and thermogravimetry (TG) were used to study the conformation, surface morphology, crystalline nature, and thermostability of the carboxymethylated EGP. The results showed that carboxymethylation did not change the triple helix structure of the EGP, but that the fundamental particles’ surface morphology was destroyed, and the crystallization area and thermal stability decreased obviously. In addition, the water solubility test and antibacterial experiment showed that the water solubility and antibacterial activity of the EGP after carboxymethylation were obviously improved, and that the water solubility of C-EGP1, C-EGP2, and C-EGP3 increased by 53.31%, 75.52%, and 80.96% respectively. The antibacterial test indicated that C-EGP3 had the best effect on *Escherichia coli* (*E. coli*) and *Staphylococcus aureus* (*S. aureus*), with minimum inhibitory concentration (MIC) values of 12.50 mg/mL and 6.25 mg/mL. The diameters of the inhibition zone of C-EGP3 on *E. coli* and *S. aureus* were 11.24 ± 0.15 mm and 12.05 ± 0.09 mm, and the antibacterial rate increased by 41.33% and 43.67%.

## 1. Introduction

*Euglena gracilis* is a kind of unicellular eukaryote belonging to *Euglena* of *Euglenophyta*. The storage polysaccharide of *Euglena gracilis* is a non-starch sticky polysaccharide linked by β-1,3 glycosidic bonds, usually referred to as paramylon [1]. Under heterotrophic conditions, *Euglena gracilis* paramylon (EGP) accumulation can reach 50~70% of the dry weight of cells [2]. Although EGP is a storage carbohydrate in *Euglena gracilis* cells, its structure differs from other β-1,3-glucans polysaccharides, which have more or less branched chains, while EGP is a strictly linear polysaccharide with high crystallinity in a natural state [3]. These linearly linked β-1,3 glycosidic bonds can form unique helical structures, which may lead to EGP showing characteristics different from those polysaccharides linked by β-1,4 glycosidic bonds [4]. As one of the available active substances in *Euglena gracilis*, EGP has attracted much attention due to its various biological activities, including removing heavy metals, balancing the intestinal environment, hepatoprotective effects, and antiviral and immunostimulatory effects [5,6,7].

However, the weak biological activity and water insolubility of many natural polysaccharides, including EGP, limit the extended application of their biological activity. It is worth noting that many researchers have changed the molecular structure and conformational characteristics of these polysaccharides through chemical modifications such as sulfation, carboxymethylation, and phosphorylation, to break these limitations and enhance their physicochemical properties and biological activity [8]. Nakashima, et al. [6], reported that the sulfated derivatives of EGP could significantly inhibit the pathological response of the human immunodeficiency virus (HIV-1 and HIV-2) on cultured cells, and that they inhibited HIV replication and expression of the HIV antigen. Furthermore, Russo et al. [9], proved that sonicated and alkalized EGP could significantly stimulate and increase immune cell-related factors in human lymphomonocytes. Many studies have also shown that the chemical modification of polysaccharides would change their biological activities and produce new biological activities under certain conditions [10,11]. For example, natural cellulose had no apparent biological activity and was water-insoluble, but it could inhibit tumors, and became soluble after carboxymethylated modification [12]. Carboxymethylation is the most widely used modification method of polysaccharides, having the advantages of simple operation and low cost. More importantly, carboxymethyl derivatives are safe and non-toxic [13]. The principle of carboxymethylation is to change polysaccharides’ structure and functional properties by introducing new groups to replace the hydroxyl groups on the polysaccharide chain (Figure 1). Other studies have also shown that the carboxymethylated polysaccharide has more vigorous biological activities, such as antitumor, antioxidation, and immune regulation. Its potential mechanism is that carboxymethyl groups can effectively improve structural characteristics, such as solubility and solution conformation of polysaccharides [14,15]. 

As an active polysaccharide in *Euglena gracilis*, EGP has been proved to have antibacterial activity, which can inhibit the growth of *E. coli* and *S. aureus* [16]. However, the antibacterial effect of EGP is not very good, due to its high crystallinity and water insolubility. Therefore, in order to change this deficiency, the main aim of this study was to change the functional properties of EGP by introducing new reactive groups through chemical modification. Carboxymethylation modification is a safe and green method, and there have been few reports on the antibacterial activity of carboxymethylated EGP. We prepared carboxymethylated EGP with different degrees of substitution, by changing the reaction conditions to improve the antibacterial activity of EGP. We also explored the changes in structural and functional characteristics after carboxymethylation. The inhibitory effect of carboxymethylated EGP on *E. coli* and *S. aureus* was evaluated. These research results could provide a theoretical reference for applying EGP in the food and pharmaceutical fields.

## 2. Materials and Methods

### 2.1. Materials and Chemicals

The *Euglena gracilis* powder was from the Guangyu Biological Technology Co. (Shanghai, China); the *E. coli* and *S. aureus* were from the School of Food Sciences and Engineering, Qilu University of Technology (Jinan, China). The NaOH, chloroacetic acid and other analytical reagents were from Sinopharm (Beijing, China).

### 2.2. Preparation of EGP

The *Euglena gracilis* powder was refluxed with petroleum ether (1: 10, *w*/*v*) in an automatic Soxhlet extractor (Naai, Shanghai, China) for 3 h, and air-dried to obtain defatted and decolorized *Euglena gracilis* powder. In accordance with the previous method [17], *Euglena gracilis* powder was suspended in 10 g/L sodium dodecyl sulfate (SDS), mixed evenly. This mixture was allowed to react in a water bath at 50 °C for 2 h, and centrifuged for 10 min at 6000 rpm. The supernatant was then removed. The next step was to add 1.5 g/L SDS solution to the precipitate, which was then centrifuged for 10 min at the same rotating speed. The supernatant was then poured out, and the operation was repeated twice. Finally, the precipitate was washed with distilled water repeatedly and freeze-dried to obtain EGP.

### 2.3. Preparation of Carboxymethylated EGP (C-EGP)

The principle of carboxymethylation is that –OH of polysaccharide is deprotonated to form alkoxide, and then –CH_2_COONa is introduced between chloroacetic acid and polysaccharide alkoxide [13]. The solvent and water methods are often used for the carboxymethylation of polysaccharides. In this experiment, the solvent method was used to modify polysaccharides, in accordance with a previous report [18], with some modifications.

Firstly, the EGP (1 g) was suspended in 30 mL of isopropyl alcohol. The mixture was stirred and swelled for 0.5 h with a magnetic stirrer (IKA, Staufen, Germany) at room temperature, following which, 3 mL 30% NaOH solution was added, to complete the alkalization process. A solid etherification reagent chloroacetic acid (1 g) was added to the reaction bottle, and stirred for 0.5 h. The temperature was set at 60 °C, and kept constant for 1.5 h to complete the etherification process. After the reaction, the precipitate—after vacuum filtration—was washed 3 times with absolute ethyl alcohol, and then dried to a constant weight in a 50 °C oven, to obtain C-EGP1. A similar procedure was used to produce C-EGP2, except that the additional amount of chloroacetic acid was 2 g. In addition, C-EGP3 was obtained in the etherification time for 50 °C.

### 2.4. Characterization

#### 2.4.1. Determination of the DS

The DS is considered an essential factor influencing polysaccharides’ structure and biological activity, and is usually determined by the neutralization titration method [19]. Firstly, C-EGP1, C-EGP2, and C-EGP3 were dried, weighed, and dissolved in a 0.1 mol/L HCl solution. After sufficient shaking, the mixture was titrated with 0.1 mol/L NaOH standard solution, and the V_1_ and V_2_ of the NaOH solution consumed at pH 2.1 and 4.3, respectively, were recorded. The DS could be calculated by equations 1, 2 [20]:(1)$$\mathrm{DS}=\frac{0.023A}{1-0.058A}$$
(2)$$A=\frac{\left(V_{2}-V_{1}\right)\times C}{m}$$

V_1_: the volume of the NaOH solution consumed (mL) when pH was 2.1; V_2_: the volume of the NaOH solution (mL) when pH was 4.3; C: NaOH solution concentration (mol/L); m: the mass of the EGP (mg).

#### 2.4.2. Component Analysis of the EGP and Its Modified Derivatives

The total sugar content of the samples was evaluated through the calibration curve of standard glucose by the phenol-sulfuric acid method [21]. The carbazole method determined the uronic acid content, and galacturonic acid (Macklin, Shanghai, China) was used as the standard [22]. Coomassie brilliant blue method was used to determine the content of protein.

#### 2.4.3. Determination of Molecular Weight

The molecular weight (Mw) was determined by gel permeation chromatography (Agilent, Palo Alto, Santa Clara, CA, USA) equipped with a differential refractive index detector and a PL gel Olexis column. The test conditions were described as follows, in accordance with the method of Wang [23], with minor modification: the column temperature was run at 30 °C, the samples (100 μL) were passed through a 0.5 μm nylon membrane and injected into the GPC, and the mobile phase was dimethyl sulfoxide at a constant flow rate of 1.0 mL/min. The Mw of samples was estimated based on the dextran of different Mw.

#### 2.4.4. FTIR Spectroscopy

An FTIR spectrophotometer (Thermo Scientific, Waltham, MA, USA) was used to analyze the infrared spectrum of the samples in the range of 500 cm^−1^ to 4000 cm^−1^ with a resolution of 4 cm^−1^. The test method was obtained from a previous report [24], with some modifications. Briefly, a small amount of dried sample powder was placed on a diamond plate for infrared transmission spectroscopy.

#### 2.4.5. NMR Spectroscopy Measurements

The ^13^C NMR spectra of the samples were measured with a Bruker Avance 400 system (Bruker, Karlsruhe, Germany), using tetramethylsilane (TMS) as the internal standard, and the chemical shifts (δ) were reported in ppm. EGP (30 mg) was dissolved in a 5 mm NMR sample tube with deuterated dimethylsulfoxide (DMSO-d6, 1.2 mL), and water-soluble C-EGPs (30 mg) were dissolved with D_2_O (1.2 mL). The spectra were scanned 1024 times at room temperature, and data were processed using the MestReNova11 program (Mestrelab Research, Santiago de Compostela, Spain).

#### 2.4.6. Congo Red Experiment

Polysaccharides with triple-helical chain conformation can form complexes with Congo red (CR), and the maximum wavelength of the complexes is redshifted, compared with CR. Therefore, CR can be used to detect whether the triple helix structure of polysaccharides is destroyed. In accordance with the description by Hou [25], the CR test method was as follows: 2 mg CR was dissolved in 100 mL of water, to prepare the CR solution. Next, 10 mg of paramylon powder was dissolved in 1.0 mol/L NaOH solution, which was then diluted with distilled water, until the NaOH concentration was 0.1~0.5 mol/L. Equal volumes of CR solution with NaOH aqueous solution containing polysaccharides were mixed well, and distilled water was used as control. The mixture was immediately scanned with a Shimadzu UV-2500 spectrophotometer (Shimadzu, Kyoto, Japan) in the wavelength range of 200 nm~800 nm, and the maximum absorption wavelength was determined.

#### 2.4.7. Morphological Characteristics

The microstructures of the samples were observed using an SEM (TESCAN, Brno, Czech) of 15 kV, in accordance with a previous report [26], with some modifications. The samples were spread onto the surface of double-sided carbon adhesive tape, and pasted on an aluminum micrometer sheet. All samples were then covered with a gold film (20 nm) in argon.

#### 2.4.8. Crystalline Characteristics

The crystallinity of the samples was determined using an XRD diffractometer (Smart Lab SE, Tokyo, Japan), in accordance with the description of Gui [27], with minor modification. Diffractograms were performed for a scattering angle of 5~40° (2θ); the scanning rate was 2°/min, and a step width was set as 0.05°. The diffractograms of the samples were plotted by Origin software.

#### 2.4.9. TG Analysis

The thermal degradation temperatures of the samples were evaluated using a thermogravimetric analyzer (Netzsch, Bavaria, Germany). In accordance with the method of Motonari [28], with minor modification, the experiment proceeded in a nitrogen atmosphere at a purge rate of 20 mL/min. Thermograms of samples were obtained between 30 °C and 750 °C, at a heating rate of 20 °C/min.

### 2.5. Water Solubility (WS) Test

The WS of the samples was determined according to the method described by Sun [29], with a slight modification. The dry samples (500 mg) were suspended with 20 mL of distilled water in 50 mL centrifuge tubes, which were agitated for 5 min using a vortex mixer. The samples stood at room temperature for 20 min. The samples were centrifuged at 6000 rpm for 10 min, and the supernatant was dried at 108 °C until it was a constant weight. The WS was calculated by Equation (3):(3)$$\mathrm{WS}\left(\%\right)=\frac{m_{1}}{m_{2}}\times 100$$ where m_1_ was the mass of the original sample (mg), and m_2_ was the mass of supernatant after drying (mg).

### 2.6. Antibacterial Activities

#### 2.6.1. Inhibition Zone Experiment

Antibacterial activity of the samples against *E. coli* and *S. aureus* was evaluated using the Oxford cup method [30]. Two bacteria were cultured in a nutrient broth medium at 37 °C for 12 h. The cultures of *E. coli* and *S. aureus* containing 10^6^ CFU/mL were prepared by diluting with sterilized physiological saline. The sterile Oxford cups were placed on the Petri dishes, and the agar was poured into the petri dish. The diluted bacterial suspension (100 μL) was spread on the nutrient agar medium after the sterile Oxford cups were removed, and the sample solutions (30 mg/mL) were separately added to the resulting well. Sterilized physiological saline and Ampicillin (100 μg/mL, Macklin, Shanghai, China) were used as the blank and positive control, respectively. Afterwards, the bacteria were incubated for 24 h at 37 °C, to measure the inhibition zones using a caliper.

#### 2.6.2. MIC Assay

The minimum inhibitory concentration (MIC) was defined as the lowest concentration of inhibiting-tested microorganisms’ growth [31]. The sample with the highest antibacterial effect was selected for the MIC test in this experiment. The bacteria were cultured by streaking on the culture medium, and whether there was bacterial growth or not was taken as the judgment standard. The samples of serial two-fold dilutions, with concentrations of 100, 50, 25, 12.5, 6.25, 3.13, and 1.56 mg/mL, were used for this test, and the lowest concentration that had no bacterial growth was taken as the MIC value of the sample. Sterilized physiological saline was used as the blank control, and all the media were incubated for 24 h at 37 °C.

#### 2.6.3. Dynamic Antibacterial Activity 

Bacterium suspensions (10 μL) and 1 mL samples (40 mg/mL) were added into test tubes containing a 9 mL nutrient medium, which were placed in an intelligent constant temperature culture oscillator (Honour, Tianjin, China) at 37 °C for 12 h. During this period, *E. coli* or *S. aureus* culture solutions were taken at 0, 2, 4, 6, 8, 10, and 12 h, respectively, and the growth curves were drawn by measuring the bacteria’s OD value.

### 2.7. Statistical Analysis

The data were shown as means ± standard deviations (SD) of three replicated determinations. Statistical analysis and analysis of variance were performed using IBM SPSS Statistics 21 software (IBM, New York, NY, USA).

## 3. Result and Discussion

### 3.1. Chemical Composition, DS and Mw Analysis

The chemical composition, the DS, and the Mw analysis of the samples are shown in Table 1. In this study, the different conditions for preparing the C-EGPs led to a different DS for each: the DS of C-EGP1, C-EGP2, and C-EGP3 was 0.14, 0.55, and 0.78, respectively, which demonstrated that carboxymethylation was successful. 

The Mw was an important factor affecting the biological activity of the polysaccharides. In accordance with the standard curve of the dextran standard, the Mw of the EGP, C-EGP1, C-EGP2, and C-EGP3 was 1.094 × 10^5^, 1.153 × 10^5^, 1.330 × 10^5^, and 1.482 × 10^5^ Da, respectively. We found that the Mw of the C-EGPs was increased, compared to that of the EGP, and that this was mainly caused by the substitution of hydroxyl by carboxymethyl groups [32]. However, some polysaccharides of Mw may have decreased due to acid hydrolysis during chemical modification [33]. Table 1 also showed significant differences in the chemical composition of samples observed after the structure modification. The protein content of all the samples was less than 1%, indicating that protein was removed effectively. The total sugar content of the samples was 55.06% (EGP), 44.72% (C-EGP1), 42.39% (C-EGP2), and 38.91% (C-EGP3). The total sugar content of the C-EGPs was significantly lower than the EGP; this may have been caused by degradation [34]. A previous study also reported that adding carboxymethyl functional groups decreased total sugar content [29]. The uronic acid content of the EGP was significantly higher than that of the C-EGPs, which was related to the β-elimination reaction in the carboxymethylation stage under alkaline conditions [35]. However, the total sugar content and uronic acid of the modified corn silk polysaccharides were increased significantly, while the protein contents were relatively reduced in another study [36]. It could be seen that the structural modification had a significant change in polysaccharides, but the results were not consistent.

### 3.2. FTIR Spectroscopy Analysis

The FTIR spectra of the EGP and its derivatives are depicted in Figure 2. All the samples showed the characteristic absorption peaks of the polysaccharides. The vast peaks between 3000 cm^−1^ and 3600 cm^−1^ represented the vibrational stretches of –OH groups; the height around 2910 cm^−1^ was C–H stretching in CH_2_ and CH_3_ radicals [37]; the peak at 1036 cm^−1^ was attributed to the ether ring (C–O–C) stretching vibration peak on the sugar ring, and 889 cm^−1^ was the β-glycosidic bond absorption peak [38]. The FTIR spectra of the C-EGPs (Figure 2b–d) showed similar peaks to EGP (Figure 2a), but the β-glycosidic bond absorption peak disappeared in the spectrum, and three additional characteristic absorption peaks appeared at 1596 cm^−1^, 1417 cm^−1^, and 1312 cm^−1^, which were related to the stretching vibration of –COO, C–C, and the symmetric stretching of C–H [39]. These proved that the C-EGPs had been successfully carboxymethylated. Similar results have been reported, that the carboxymethyl cellulose-based films showed three new peaks at 1590 cm^−1^,1410 cm^−1^, and 1320 cm^−1^ in the infrared spectrum after carboxymethylation modification [40].

### 3.3. NMR Analysis

In order to further understand whether carboxymethylation was successful or not, we performed a ^13^C NMR measurement on the C-EGPs (Figure 3). The ^13^C NMR spectrum of the natural EGP (Figure 3a) showed six different glucose-derived signals at 60.9, 68.4, 72.9, 76.4, 86.3, and 103.1 ppm, which belonged to C6, C4, C2, C5, C3, and C1 respectively, which was consistent with a previous report [41]. For the ^13^C NMR spectrum of the C-EGPs, it could be observed that new signals with different intensities appeared at about 180 ppm, and the appearance of these new signals was due to carbonyl carbon of the carboxymethyl group, which indicated the success of carboxymethylation [18]. Due to the different degrees of carboxymethylation, C-EGP3 (Figure 3d) showed two signals at about 180 ppm, and the signal intensity was more substantial than that of C-EGP2 (Figure 3c) and C-EGP1 (Figure 3b). In addition, we could see that the signal intensity of C6 of C-EGP3 and C4 of C-EGP2 were more potent than that of the EGP from Figure 3. Shibakami, et al. [42], carried out a ^13^C spectrum of carboxylic acid-bearing polysaccharide nanofibers made from euglenoid β-1,3-glucan and succinic anhydride. The results showed the appearance of ^13^C signals at 31.0, 32.5 ppm in the lower resonance frequency, and 176.2, 181.6 ppm in the higher resonance frequency, ascribed to the succinyl ester carbonyl and methylene carbon atoms, respectively. Furthermore, the succinylated paramylon spectrum showed that the intensities of the C2, C4, C5, and C6 signals were higher than those of C1 and C3, which was consistent with the results of C6 and C4 of the C-EGPs in this study. However, there were no new signals in the lower resonance frequency in the spectra of the C-EGPs, which may be due to the different substitution degrees, and uneven distribution of carboxymethyl groups within three hydroxyls of one glucose unit.

### 3.4. CR Analysis

The carboxymethyl groups replaced the –OH groups of the polysaccharides chain after successful carboxymethylation, which may have led to the changes in conformations of the polysaccharides [43]. The λ_max_ change of the EGP and its derivatives, under different concentrations of sodium hydroxide, is shown in Figure 4. We found that the λ_max_ of the EGP solution was redshifted, which indicated that the EGP had a triple helix structure. The λ_max_ of the EGP decreased sharply when the NaOH concentration reached 0.2 mol/L, which may have been due to the destruction of the helical structure of the EGP in the NaOH solution. The C-EGPs also showed a redshift, which indicated that the modification did not destroy the conformation of the polysaccharides. However, Liu [30] found that carboxymethylation of polysaccharides (cmCVP-1S_9_ and cmCVP-1S_10_) did not show a redshift. It was speculated that the DS of cmCVP-1S_9_ and cmCVP-1S_10_ were much higher than the C-EGPs through comparative analysis, but this did not rule out the difference between the polysaccharides. Notably, we also found that the samples showed an enormous λ_max_ value with the increase of the DS, which meant that a proper amount of carboxylic acid groups enabled the polysaccharides to sustain a relatively stable triple helix structure in an alkaline environment.

### 3.5. Surface Morphology Analysis

SEM was used to investigate the microstructure and surface characteristics of the samples, and the SEM images of sample particles for 5.0 k and 10.0 k magnifications are given in Figure 5. The size and shape of the EGP and the C-EGPs were significantly different in the SEM images. The EGP particles were oval, with a uniform and smooth surface (Figure 5(A1),(A2)). The surface of C-EGP1 was cracked, to varying degrees, and the shape of the sample changed (Figure 5(B1),(B2)). Liu [44] also found that polysaccharides after carboxymethylation treatment showed irregular fragments, while polysaccharides before treatment had a smooth appearance of different sizes. In addition, the shape change, and the phenomenon of the particles breaking up in the C-EGPs, became more distinct with the increase of the DS. The most intuitive manifestation was that C-EGP3 (Figure 5(D1),(D2)) was broken, to form a rougher surface and smaller particles than C-EGP2 (Figure 5(C1),(C2)). This may have been due to the change of the polysaccharide molecular structure. At the same time, the SEM results also verified the results of XRD: that the degree of carboxymethylation affected the degree of polysaccharide crystallization. It is illuminating that the improvements in crystallinity were positively correlated to the density of the surface structure: the looser the surface structure of the polysaccharide, the smaller the degree of crystallinity.

### 3.6. XRD Analysis

Crystallinity and crystal type reflected the samples’ molecular structures and characteristics [45]. The XRD patterns of the EGP and the C-EGPs are displayed in Figure 6. As shown in Figure 6a, the EGP had a sharp and narrow peak at 6.8°, 19.3°, 20.5°, and 23.9°, and three small peaks at 11.8°,13.8°, and 16.6°. This was consistent with the paramylon particles showing sharp peaks at 6.8°, 11.9°, 13.9°, 16.6°, 19.3°, 20.5°, and 23.9°, reported by Shibakami, et al. [18]. The crystal peaks of C-EGP1 at 6.9°, 19.5°, 20.5°, and 23.9° reduced obviously, and three small peaks around 12°,14°, and 17° disappeared (Figure 6b). For tC-EGP2 and C-EGP3, all of the height peaks disappeared. In other words, the XRD curves of C-EGP2 and C-EGP3 showed a broad peak representing an amorphous halo around 20.0°, which meant that the sample changed from a highly crystalline state to an amorphous state. 

By analyzing the crystalline state of C-EGP1, C-EGP2, and C-EGP3, it was found that the crystallinity decreased with the increase of the DS, which meant that the ordered initial arrangement of the EGP was disturbed. In addition, the two solid derivatives (Figure 6c,d) showed one sharp peak around 30°, due to residual sodium chloride.

### 3.7. Thermogravimetric Analysis

A thermogravimetric (TG) analysis, and the differential thermogravimetric (DTG) curves, are shown in Figure 7. The degradation temperatures and weight losses of the samples are presented in Table 2. Each step of thermal degradation corresponded to a specific degradation behavior of polysaccharides [46]. The weight loss could be divided into two stages over the test range of 30~750 °C, corresponding to the two degradation peaks in the DTG curves. The initial weight loss stage at 30~150 °C was usually attributed to the loss of water [47]. The primary reaction was observed around 150~750 °C, corresponding to the destruction of the samples’ hydrogen bond or sugar decomposition [48]. The initial degradation temperature of the C-EGPs was lower than that of the EGP (324.88 °C), indicating that the thermal stability of the EGP was debased by carboxymethylation modification. The results corresponded to the exposure of a larger surface area of C-EGPs in the SEM images (Figure 5), and the loose crystal structure in Figure 6. Furthermore, the thermal degradation temperature of C-EGP1, C-EGP2, and C-EGP3 decreased, indicating that a higher DS would decrease the material’s thermal stability.

### 3.8. Water Solubility Analysis

The water solubility test results (Table 3) showed that the EGP was dispersed in water, closely related to its high molecular weight and crystallinity [6]. In addition, it could be seen that the water solubility of the C-EGPs was increased, which was attributed to the excellent hydrophilicity of the carboxymethyl group. The water solubility of C-EGP1, C-EGP2, and C-EGP3 increased sequentially with the DS, and the water solubility of C-EGP3 reached 80.96%. This was due to the hydrogen bonds between the carboxymethyl groups, and water increase with the increase of the number of carboxymethyl groups, which reduced the formation of intramolecular hydrogen bonds and improved the water solubility [14].

### 3.9. Antibacterial Activity

#### 3.9.1. Bacterial Inhibition Zone Essay Analysis

The results of the inhibition zone of the EGP and its derivatives against *E. coli* and *S. aureus* are shown in Table 4. The results showed that the EGP and its derivatives displayed antibacterial activity against the tested strains, and that the C-EGPs had a more substantial inhibitory effect. Meanwhile, in recent years, many pieces of research have also shown that carboxymethylation can promote the antibacterial effect of polysaccharides. Erum, et al. [49], found that carboxymethyl arabinoxylan enhanced antibacterial activity against *S. epidermidis*, *S. aureus*, and *B. subtilis*. Song, et al. [50], also reported that carboxymethylation could form bubbles and microspheres by inducing membrane permeability and structural damage in *S. aureus* cells, to improve the antibacterial activity of β-Glucan. It is worth noting that the antibacterial activity of C-EGP3 was higher than C-EGP1 and C-EGP2, which revealed that the DS was essential in inhibiting bacteria growth. Liu, et al. [30], also found that appropriate DS helped improve the antibacterial activities of carboxymethylated polysaccharides from *Catathelasma ventricosum*, but that high DS would destroy the triple helix conformation of the polysaccharides, leading to a weak bacteriostatic effect. Similarly, this study’s CR test results (Figure 4) showed that the triple helix structure of the C-EGPs with three different DS had not been destroyed; thus, they showed good antibacterial activity.

#### 3.9.2. MIC Analysis

The MIC assay demonstrated the sequence of susceptibility of the samples. In this study, the MIC values of C-EGP3 against the tested strains are shown in Table 5. The MIC of C-EGP3 against *S. aureus* (6.25 mg/mL) was lower than against *E. coli* (12.5 mg/mL). This proved that *S. aureus* was more sensitive to C-EGP3. Similarly, Sudipta, et al. [51], found that *E. coli* bacteria were highly resistant, due to their outer membranes (OM). In addition, the results regarding the MIC sensitivity order of strains were in good agreement with those of the inhibition zone test, and consistent with the research results of Shao [52]. When the concentration of C-EGP3 was 12.5 mg/mL, C-EGP3 had an inhibitory effect on the two bacterial species. According to the results, carboxymethylated C-EGPs can be considered potential antibacterial agents. 

#### 3.9.3. Dynamic Antibacterial Activity Analysis

In order to better understand the dynamic antibacterial effect of EGP, and its derivatives, on the growth of the tested strains, we detected and analyzed the OD values in the bacterial solution every two h, to quantitatively obtain the antibacterial effect. 

Figure 8 shows that the inhibition rates of C-EGP1, C-EGP2, and C-EGP3 against *E. coli* were increased by 14.99%, 24.65%, and 41.33%, respectively, while the inhibition rates against *S. aureus* were increased by 15.55%, 30.71%, and 43.67%, respectively. This indicated that C-EGP3 showed a better antibacterial effect in the whole growth process, and that the antibacterial effect of all the samples against *S. aureus* was stronger than against *E. coli*. The inhibitory effect of the EGP against test strains was more evident in the dynamic assay process than in the inhibition zone experiment, which may be due to the poor diffusion effect in agar caused by the water insolubility of the EGP, whereas the EGP dispersed better in the culture medium liquid. However, the carboxymethylated polysaccharides (C-EGP1, C-EGP2, C-EGP3) did not appear in this phenomenon, due to their excellent water solubility. Meanwhile, Ren et al. [53], also found that the result of the bacterial inhibition zone test was affected by polysaccharides’ solubility and diffusion ability.

## 4. Conclusions

In this study, EGP was extracted from *Euglena gracilis*, and three carboxymethylated polysaccharides (C-EGPs), each with a different DS, were prepared by solvent method. Meanwhile, the inhibitory effect on *E. coli* and *S. aureus* was investigated. In the FTIR spectrum, the appearance of the characteristic peak 1596 cm^−1^, 1417 cm^−1^ and 1312 cm^−1^ indicated that the carboxymethyl group was successfully introduced into the molecular structure of the EGP, while the NMR spectrum showed that the carboxymethyl group was substituted at the C4 and C6 positions. In addition, we found that the content of total sugar and uronic acid was decreased through the determination of the basic components of the samples, which meant that the carboxymethylation process was accompanied by the degradation of polysaccharides. The microstructural analysis indicated that the EGP particles were broken, and that the surface became rough after carboxymethylation modification. Further, based on XRD spectroscopy and TG analysis, the crystalline region and thermal stability of the C-EGPs decreased with the increase of the DS, and this also indicated that the lower the crystallinity of the material, the lower the thermal stability. The water solubility of the C-EGPs was significantly improved: the solubility of C-EGP3 reached 80.96%, which would significantly expand the application range of EGPs. Through the antibacterial test, we found that the C-EGPs had enhanced antibacterial activity against *E. coli* and *S. aureus*, and that the inhibitory effect on *S. aureus* was more significant. It is also worth noting that C-EGP3 showed stronger antibacterial effect relative to other samples, and that the inhibition rates against *E. coli* and *S. aureus* reached 41.33% and 43.67%, which highlighted that the carboxymethyl group was the key factor responsible for increasing the antibacterial activity. In summary, this study demonstrates that C-EGPs can be used as a potential antibacterial ingredient in functional food, packaging, and medical industries, and also shows that the structure of polysaccharides plays an important role, affecting their functional properties and biological activities.

## Figures and Tables

**Figure 1 polymers-14-03022-f001:**
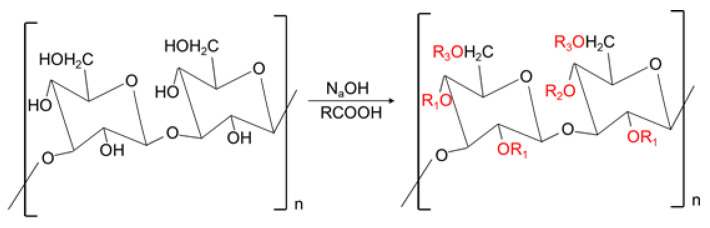
Schematic diagram of carboxymethylation principle.

**Figure 2 polymers-14-03022-f002:**
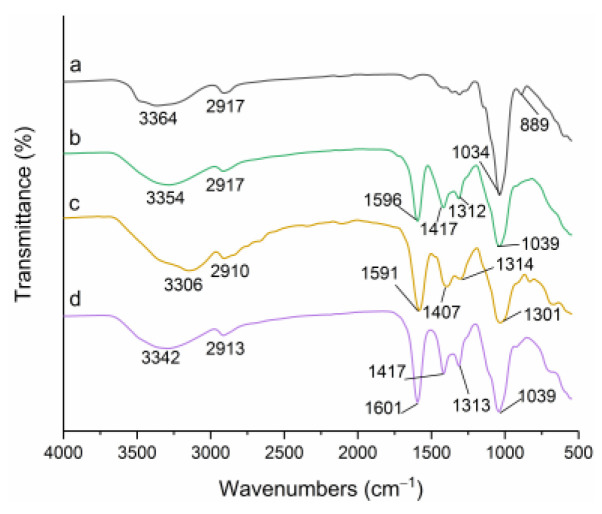
FTIR spectra of the EGP and its derivatives: **a**, **b**, **c**, and **d** represent EGP, C-EGP1, C-EGP2, and C-EGP3.

**Figure 3 polymers-14-03022-f003:**
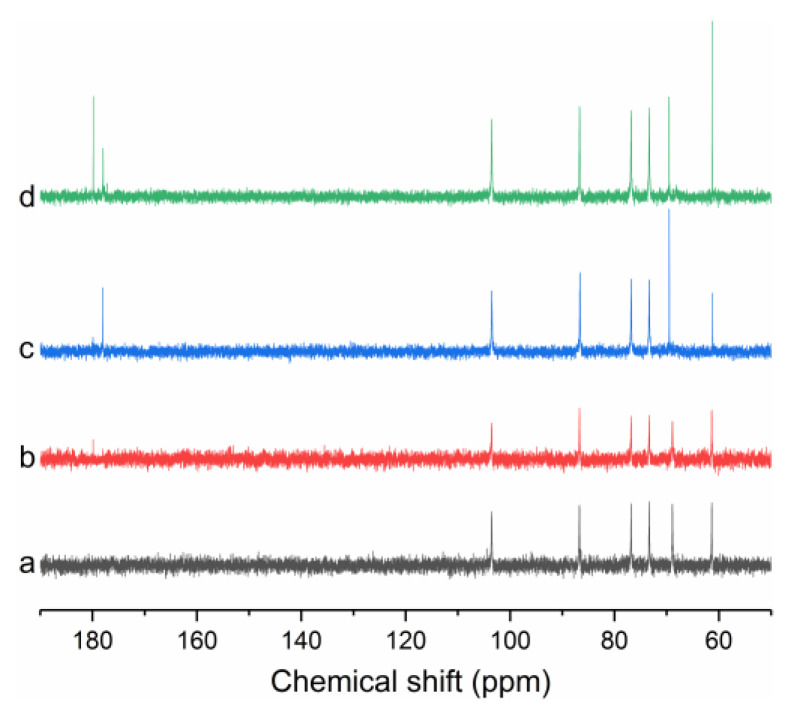
(**a**) ^13^C NMR spectrum of EGP; (**b**) C-EGP1; (**c**) C-EGP2; (**d**) C-EGP3. The EGP was dissolved with DMSO-d6; the C-EGPs were dissolved with D_2_O.

**Figure 4 polymers-14-03022-f004:**
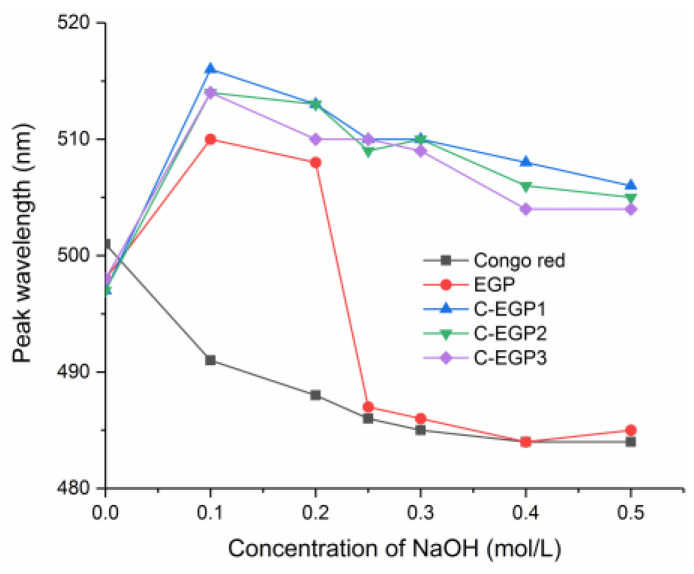
Change of samples’ λ_max_ at various NaOH concentrations.

**Figure 5 polymers-14-03022-f005:**
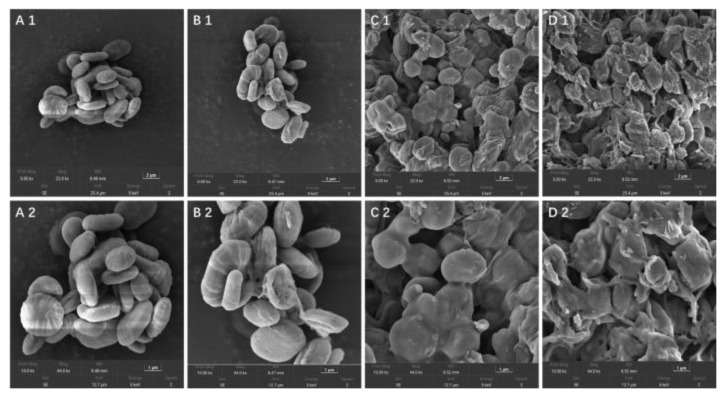
SEM micrographs of the EGP (**A1**,**A2**), C-EGP1 (**B1**,**B2**), C-EGP2 (**C1**,**C2**), and C-EGP3 (**D1**,**D2**).

**Figure 6 polymers-14-03022-f006:**
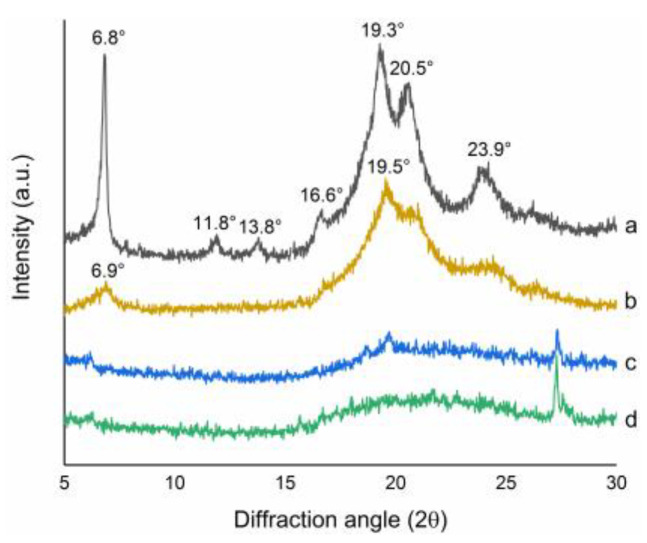
X-ray diffractions of the EGP and its derivatives: (**a**) EGP; (**b**) C-EGP1; (**c**) C-EGP2; (**d**) C-EGP3.

**Figure 7 polymers-14-03022-f007:**
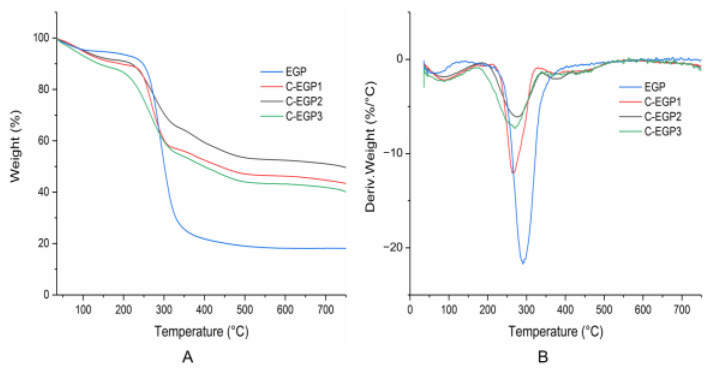
TG (**A**) and DTG (**B**) curves of samples.

**Figure 8 polymers-14-03022-f008:**
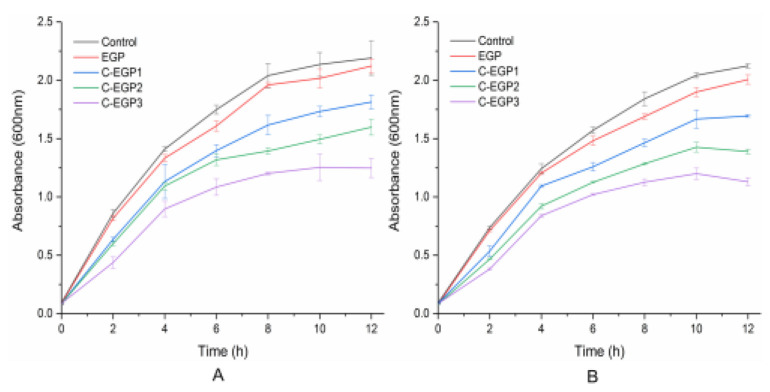
The dynamic antibacterial effect of the EGP and its derivatives against *E. coli* (**A**) and *S. aureus* (**B**).

**Table 1 polymers-14-03022-t001:** Chemical Composition, DS, and Mw of EGP and C-EGPs.

Samples	EGP	C-EGP1	C-EGP2	C-EGP3
Total sugar (%)	55.06 ± 0.992 ^a^	44.72 ± 0.121 ^b^	42.39 ± 0.866 ^c^	38.91 ± 0.651 ^d^
Uronic acid (%)	19.76 ± 1.348 ^a^	17.67 ± 1.332 ^b^	16.25 ± 0.265 ^b^	13.08 ± 0.291 ^c^
Protein (%)	0.74 ± 0.045 ^a^	0.72 ± 0.021 ^a^	0.50 ± 0.067 ^b^	0.37 ± 0.047 ^c^
Mw (×10^5^)	1.094	1.153	1.330	1.482
DS	–	0.14 ± 0.005 ^c^	0.55 ± 0.01 ^b^	0.78 ± 0.015 ^a^

Values are mean ± SD, n = 3. Values in the same column with different letters are significantly different (*p* < 0.05).

**Table 2 polymers-14-03022-t002:** Thermogravimetric Analysis of Samples.

Sample	First-Stage Weight Loss (30~150 °C)		Second-Stage Weight Loss (150~750 °C)		Residue (%)
	T_d1_ (°C)	Δw_1_ (%)	T_d2_ (°C)	Δw_2_ (%)	
EGP	38.44 ± 0.52 ^b^	6.1 ± 1.26 ^d^	324.88 ± 2.81 ^a^	84.81 ± 1.81 ^a^	9.08 ± 0.60 ^d^
C-EGP1	44.95 ± 0.63 ^a^	12.9 ± 2.28 ^bc^	244.72 ± 0.84 ^b^	46.19 ± 0.56 ^b^	40.25 ± 0.58 ^b^
C-EGP2	44.07 ± 0.79 ^a^	15.07 ± 1.5 ^b^	231.53 ± 1.05 ^c^	40.33 ± 0.49 ^c^	44.6 ± 1.034 ^a^
C-EGP3	37.15 ± 0.7 ^b^	19.85 ± 2.78 ^a^	220.50 ± 1.06 ^d^	48.12 ± 1.07 ^b^	32.03 ± 1.72 ^c^

Data are expressed as means ± SD (n = 3). Values in the same column with different letters are significantly different (*p* < 0.05).

**Table 3 polymers-14-03022-t003:** Water Solubility Analysis of Samples.

Samples	EGP	C-EGP1	C-EGP2	C-EGP3
Water solubility (%)	0.52 ± 0.27 ^d^	53.31 ± 1.03 ^c^	75.52 ± 2.95 ^b^	80.96 ± 1.45 ^a^

Data are expressed as means ± SD (n = 3). Values in the same column with different letters are significantly different (*p* < 0.05).

**Table 4 polymers-14-03022-t004:** Antibacterial Activity of EGP and its Derivatives.

Item (30 mg/mL)	Diameters of Inhibition Zone (mm)	
	*E. coli*	*S. aureus*
EGP	6.08 ± 0.100 ^d^	6.08 ± 0.055 ^d^
C-EGP1	8.01 ± 0.121 ^c^	10.10 ± 0.187 ^c^
C-EGP2	9.30 ± 0.239 ^b^	11.19 ± 0.137 ^b^
C-EGP3	11.24 ± 0.151 ^a^	12.05 ± 0.092 ^a^
Physiological saline	6.00	6.00
Ampicillin *	18.23 ± 0.226	20.39 ± 0.539

Each value is expressed as the mean ± SD of triplicate analysis. Means with different letters within a column are significantly different (*p* < 0.05). * Ampicillin at a concentration of 100 μg/mL.

**Table 5 polymers-14-03022-t005:** The Minimum Inhibitory Concentration of C-EGP3.

Concentration (mg/mL)	Growth Status	
	*E. coli*	*S. aureus*
100	–	–
50	–	–
25	–	–
12.5	–	–
6.25	+	–
3.13	+ +	+
1.56	+ +	+ +
Physiological saline	+ +	+ +

Note: “–” no activity; “+” means growing bacteria; “+ +” means exuberant growth.

## Data Availability

Not applicable.

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
