# Peer review of "Characterization and Antibacterial Activities of Carboxymethylated Paramylon from Euglena gracilis"

_polymers, 2022, doi:10.3390/polym14153022_

Round 1
Reviewer 1 Report
Characterization and antibacterial activities of carboxymethyl- 2 ated paramylon from Euglena gracilis is a good MS but there are some issues to be addressed.
A list of my comments are listed below:
1. Aim of the work should be revised to clarify its novelty.
2. detailed information about NaOH and chloroacetic acid should be added to the material section.
3. FTIR Fig. quality is very poor and should be improved.
4. SEM images need more discussion and the scale bar should be clearly shown.
5. The XRD peaks of the present study are differ from the reported work? explain
6. TG and DTG curves of samples are of bad quality and should be improved
7. Fig 7 quality should be improved.
8. Conclusion should be rewritten to focus on the article outcomes.
9. English should be revised carefully.
Author Response
Point 1: Aim of the work should be revised to clarify its novelty.
Response 1: The aim of the work have been revised in introduction of the manuscript.
Point 2: detailed information about NaOH and chloroacetic acid should be added to the material section.
Response 2: The detailed information about NaOH and chloroacetic acid have been added to the material section.
Point 3: FTIR Fig. quality is very poor and should be improved.
Response 3: The FTIR Fig. have been revised in the manuscript.
Point 4: SEM images need more discussion and the scale bar should be clearly shown.
Response 4: The more discussion have been added in section 3.5 of the manuscript, and the scale bar also have been added the SEM images.
Point 5: The XRD peaks of the present study are differ from the reported work? Explain
Response 5: The XRD peaks in this study are consistent with previous reports, and the misunderstanding due to our rounding of the data. Meanwhile, we have reorganized the data, and you can see the detailed modification in section 3.4.
Point 6: TG and DTG curves of samples are of bad quality and should be improved
Response 6: The TG and DTG curves of samples have been revised.
Point 7: Fig 7 quality should be improved.
Response 7: The Fig 7 have been revised in the manuscript.
Point 8: Conclusion should be rewritten to focus on the article outcomes.
Response 8: The conclusion have been rewritten and marked in the manuscript.
Point 9: English should be revised carefully.
Response 9: According to your suggestion, the English of the article has been revised by native English speakers.
Reviewer 2 Report
In this submission, the author has synthesized and characterized three carboxymethylated derivatives C-EGP1, C-EGP2 and C-EGP3 with a degree of substitution 0.14, 0.55 and 0.78, respectively. The author has reported that irrespective of others, C-EGP3 showed the best antibacterial activity against the E. coli and S. aureus bacteria with MIC values of 12.50 and 6.25 mg/mL. The manuscript has been well written and could be published after the revision.
Here are the comments which need to be addressed.
1. A schematic presentation of the chemical structure of carboxymethylated paramylon could increase the clarity of presentation. So, please include this in this manuscript.
2. "CMC-based films showed three"
To increase the readability, it would be good to mention the full form of CMC.
3. Line 268: "new signals in the lower magnetic fields "
The magnetic field should be replaced by resonance frequency. Please modify it accordingly.
4. Figure 2: Please mention the solvent used for NMR in the figure caption. It would be good to have 1H NMR spectra of modified EGPs.
5. Figure 5: Y-axis’s unit should be mentioned here.
6. Line 389: “essential in inhibiting bacteria”
It should be inhibiting bacterial growth instead of bacteria. Please correct it accordingly.
7. Line 402: It would be good to mention why S. aureus is more sensitive to C-EGP3 than E. coli. Because E. coli has an outer membrane. The author further refers to the article to include the explanation.
Article title: ‘Coordination-Assisted Self-Assembled Polypeptide Nanogels to Selectively Combat Bacterial Infection’
Line 402: Also, please correct the typo of E. coli.
Author Response
Point 1: A schematic presentation of the chemical structure of carboxymethylated paramylon could increase the clarity of presentation. So, please include this in this manuscript.
Response 1: According to your suggestion, a schematic presentation of the chemical structure of carboxymethylated paramylon have been added in the manuscript, and named the diagram as Figure 1.
Point 2: "CMC-based films showed three" To increase the readability, it would be good to mention the full form of CMC.
Response 2: According to your suggestion, the "CMC-based films showed three" have been revised the "carboxymethyl cellulose-based films showed three".
Point 3: Line 268: "new signals in the lower magnetic fields "
The magnetic field should be replaced by resonance frequency. Please modify it accordingly.
Response 3: The above questions you raised have been revised and marked in the manuscript.
Point 4: Figure 2: Please mention the solvent used for NMR in the figure caption. It would be good to have 1H NMR spectra of modified EGPs.
Response 4: The solvent used for NMR have been added in the figure caption. Thank you very much for your suggestion, the 1H NMR spectra is indeed an important characterization method to prove the successful modification of C-EGPs. However, the 1H NMR spectrum will generate multiple peaks due to the influence of other protons and can not accurately specify the position of carboxymethylation substitution. So we only used 13C NMR spectra for more intuitive presenting the positions substituted.
Point 5: Figure 5: Y-axis’s unit should be mentioned here.
Response 5: The Y-axis’s unit have been revised in the Figure.
Point 6: Line 389: “essential in inhibiting bacteria”
It should be inhibiting bacterial growth instead of bacteria. Please correct it accordingly.
Response 6: “essential in inhibiting bacteria” has been changed to“essential in inhibiting bacteria growth”in the manuscript.
Point 7: Line 402: It would be good to mention why S. aureus is more sensitive to C-EGP3 than E. coli. Because E. coli has an outer membrane. The author further refers to the article to include the explanation.
Article title: ‘Coordination-Assisted Self-Assembled Polypeptide Nanogels to Selectively Combat Bacterial Infection’
Line 402: Also, please correct the typo of E. coli.
Response 7: According to your suggestion, we have explained in the manuscript why S. aureus is more sensitive than E. coli, and have added the reference you provided. The spelling of E. coli has also been revised.